# The Impact of Surgical Waiting Time on Oncological Outcomes in Patients with Upper Tract Urothelial Carcinoma Undergoing Radical Nephroureterectomy: A Systematic Review

**DOI:** 10.3390/jcm11144007

**Published:** 2022-07-11

**Authors:** Łukasz Nowak, Wojciech Krajewski, Jan Łaszkiewicz, Bartosz Małkiewicz, Joanna Chorbińska, Francesco Del Giudice, Keiichiro Mori, Marco Moschini, Krzysztof Kaliszewski, Paweł Rajwa, Ekaterina Laukhtina, Shahrokh F. Shariat, Tomasz Szydełko

**Affiliations:** 1Department of Minimally Invasive and Robotic Urology, University Center of Excellence in Urology, Wroclaw Medical University, 50-556 Wroclaw, Poland; jasieklaszkiewicz@gmail.com (J.Ł.); bmalkiew01@gmail.com (B.M.); joanna.chorbinska@gmail.com (J.C.); tomasz.szydelko1@gmail.com (T.S.); 2Department of Maternal-Infant and Urological Sciences, Policlinico Umberto I Hospital, “Sapienza” Rome University, 00185 Rome, Italy; francesco.delgiudice@uniroma1.it; 3Department of Urology, The Jikei University School of Medicine, Tokyo 105-8461, Japan; morikeiichiro29@gmail.com; 4Division of Experimental Oncology, Department of Urology, Urological Research Institute, Vita-Salute San Raffaele University, 20132 Milan, Italy; marco.moschini87@gmail.com; 5Department of General, Minimally Invasive and Endocrine Surgery, Wroclaw Medical University, 50-556 Wroclaw, Poland; krzysztof.kaliszewski@umw.edu.pl; 6Department of Urology, Comprehensive Cancer Center, Medical University of Vienna, 1090 Vienna, Austria; pawelgrajwa@gmail.com (P.R.); katyalaukhtina@gmail.com (E.L.); sfshariat@gmail.com (S.F.S.); 7Department of Urology, Medical University of Silesia, 41-800 Zabrze, Poland; 8Institute for Urology and Reproductive Health, I.M. Sechenov First Moscow State Medical University, 119435 Moscow, Russia; 9Department of Urology, Weill Cornell Medical College, New York, NY 10021, USA; 10Department of Urology, University of Texas Southwestern, Dallas, TX 75390, USA; 11Hourani Center for Applied Scientific Research, Al-Ahliyya Amman University, Amman 19328, Jordan; 12Department of Urology, Second Faculty of Medicine, Charles University, 15006 Prague, Czech Republic

**Keywords:** upper tract urothelial carcinoma, radical nephroureterectomy, delay, deferred, oncological outcomes

## Abstract

Radical nephroureterectomy (RNU) with bladder cuff excision is a standard of care in patients with high-risk upper tract urothelial carcinoma (UTUC). Although several recommendations and guidelines on the delayed treatment of urologic cancers exist, the evidence on UTUC is scarce and ambiguous. The present systematic review aimed to summarize the available evidence on the survival outcomes after deferred RNU in patients with UTUC. A systematic literature search of the three electronic databases (PubMed, Embase, and Cochrane Library) was conducted until 30 April 2022. Studies were found eligible if they reported the oncological outcomes of patients treated with deferred RNU compared to the control group, including those patients treated with RNU without delay. Primary endpoints were cancer-specific survival (CSS), overall survival (OS), and recurrence-free survival (RFS). In total, we identified seven eligible studies enrolling 5639 patients. Significant heterogeneity in the definition of “deferred RNU” was found across the included studies. Three out of five studies reporting CSS showed that deferring RNU was associated with worse CSS. Furthermore, three out of four studies reporting OS found a negative impact of delay in RNU on OS. One out of three studies reporting RFS found a negative influence of delayed RNU on RFS. While most studies reported a 3 month interval as a significant threshold for RNU delay, some subgroup analyses showed that a safe delay for RNU was less than 1 month in patients with ureteral tumors (UT) or less than 2 months in patients with hydronephrosis. In conclusion, long surgical waiting time for RNU (especially more than 3 months after UTUC diagnosis) could be considered as an important risk factor having a negative impact on oncological outcomes in patients with UTUC; however, the results of the particular studies are still inconsistent. The safe delay for RNU might be shorter in specific subsets of high-risk patients, such as those with UT and/or hydronephrosis at the time of diagnosis. High-quality additional studies are required to establish evidence for valid recommendations.

## 1. Introduction

Upper tract urothelial carcinoma (UTUC) is a rare neoplasm accounting for 5–10% of all urothelial cancers [1]. Radical nephroureterectomy (RNU) with bladder cuff excision is considered to be the treatment of choice in patients with high-risk UTUC, regardless of the primary tumor location [2]. Currently, kidney-sparing surgeries (KSS) are preferred in selected low-risk cases, as they can reduce morbidity without compromising survival endpoints [2].

The issues of surgical prioritization and establishment of recommendations regarding acceptable delays in urological procedures have been of paramount importance for the past months, due to the critical period of the COVID-19 pandemic. Although several valuable recommendations providing an overview of the risks from delayed treatment for urologic cancers exist [3,4], the evidence on UTUC remains scarce and ambiguous.

Apart from limitations of the health care systems and lowered capacity of large-volume centers due to the COVID-19 pandemic [5], multiple other elements can delay RNU. The main patient-related factor influencing the time of RNU is the presence of serious comorbidities that require alignment before surgery, as the vast majority of patients diagnosed with UTUC are elderly [6]. Moreover, some patients can be hesitant to undergo the surgery due to psychological factors. On the other hand, specific disease-related factors, such as administration of neoadjuvant chemotherapy (NAC) or performance of additional diagnostic procedures (e.g., ureteroscopy (URS)), can affect the time of RNU.

The present systematic review aimed to summarize the available evidence on the survival outcomes after deferred RNU in patients with UTUC.

## 2. Materials and Methods

This systematic review was performed according to the Preferred Reporting Items for Systematic Reviews and Meta-analysis (PRISMA) guidelines and methods outlined in the Cochrane Handbook for Systematic Reviews of Interventions [7,8]. The study protocol was registered a priori on the International Prospective Register of Systematic Reviews (PROSPERO) with the registration number CRD42022303744.

### 2.1. Search Strategy

A systematic literature search of the three electronic databases (PubMed, Embase, and Cochrane Library) was performed using the following search string: (“upper tract urothelial carcinoma” OR “upper tract urothelial cancer” OR “upper tract urothelial neoplasm” OR “upper urinary tract carcinoma” OR “upper urinary tract cancer” OR “UTUC” OR “UUTC”) AND (“nephroureterectomy” OR “RNU” OR “surgery” OR “surgical treatment” OR “operation”) AND (“delay” OR “defer” OR “deferred” OR “waiting” OR “time” OR “timing” OR “interval”). The last search was conducted on 30 April 2022. Only articles written in English (without time limitations) were considered. A cross-referenced search was additionally performed from articles selected for full-text review. Moreover, additional articles were screened from ahead-of-print articles published in various urological journals.

### 2.2. Inclusion and Exclusion Criteria

Studies were assessed for eligibility using the PICO (population, intervention, comparison, outcome) approach. The inclusion criteria were as follows:(P)opulation: Patients with UTUC who underwent RNU.(I)ntervention: Patients who underwent deferred RNU. Only studies reporting a specific cut-off defining the delay in RNU were included.(C)omparison: Patients who underwent RNU without delay.(O)utcome: The primary outcomes were cancer-specific survival (CSS) and overall survival (OS). The secondary outcome was recurrence-free survival (RFS).

The general exclusion criteria were as follows: (1) noncomparative studies—reviews, letters, editorial comments, meeting abstracts, replies from authors, case reports; (2) studies not reporting any outcome of interest.

### 2.3. Data Extraction

Data from eligible studies were independently extracted by two research authors (Ł.N. and J.L.). A standardized data extraction form was created and used to collect: study-related data (first author, publication year, journal, geographical region, study type, study duration, number of patients, reported definition of RNU delay, median time to RNU, follow up period), clinicopathological data (gender, proportion of patients with hydronephrosis, tumor location, RNU approach, pathological tumor stage, pathological tumor grade, proportion of patients with pathologically confirmed lymph node invasion (LNI), proportion of patients with concomitant carcinoma in situ (CIS), proportion of patients with positive lymphovascular invasion (LVI), and proportion of patients who received adjuvant chemotherapy (AC)), and survival data (including 5-year CSS, OS, and RFS rates, as well as their corresponding unadjusted or adjusted hazard ratios (HRs) with 95% confidence intervals (CIs)).

### 2.4. Quality Assessment and Risk of Bias

The “risk of bias” (RoB) for the selected studies was independently assessed by two review authors (Ł.N. and W.K.) according to the principles outlined in the Cochrane Handbook for Systematic Reviews and Interventions [8]. The articles were assessed in terms of allocation, sequence generation and concealment, blinding of participants, personnel and outcome assessors, completeness of outcome data, selective outcome reporting, and other sources of bias. The selected studies were also reviewed based on the adjustment for the effect of the following confounders: pathological tumor stage, pathological tumor grade, concomitant CIS, LVI, LNI, and tumor location. The risk of confounding bias was considered to be high if the confounder was not controlled for in multivariate analysis.

## 3. Results

### 3.1. Literature Search Results

The PRISMA flow chart summarizing the process of study selection was presented in Figure 1. The initial literature search identified 1258 potentially relevant references. Using literature manager software—Endnote 20 (Clarivate)—346 duplicate records were removed. After screening the titles and abstracts of identified papers, 487 and 20 articles were excluded, due to inappropriate type (e.g., review, case series, meeting abstract) and non-English language, respectively. Among the remaining 405 original studies, 389 were not relevant to the present systematic review, leaving 16 potentially eligible papers. Of the 16 full-text articles assessed for eligibility, 9 were excluded based on the predefined selection criteria.

### 3.2. Features of Included Studies

Finally, we included seven full-text studies (Table 1) [9,10,11,12,13,14,15]. Overall, the included studies enrolled 5639 patients. All articles were retrospective series, of which: three were single-center series [10,12,15], three were multi-center series [9,11,13], and one was population-based registry (data from the National Cancer Database, NCDB) [14]. One study provided data from a worldwide dataset [13], while the remaining papers included data from Asian (*n =* 3) [9,10,15], North American (*n =* 2) [12,14], and European (*n =* 1) [11] populations.

The clinical and pathological characteristics of cohorts were provided in Table 2. Most of the patients were male (54.8%) [10,11,12,13,14,15]. The proportion of non-muscle-invasive (<pT2) and muscle-invasive tumors (≥pT2) was roughly equal in four articles [11,12,13,14], while another three studies reported a higher proportion of muscle-invasive tumors [9,10,15]. Predominance of grade 3 (G3) or high-grade (HG) tumors was observed in all included studies. LNI and LVI rates ranged from 4.1% to 12.5% and 12% to 30.8%, respectively. The proportion of patients receiving AC ranged from 10 to 31.2%. All studies except one [12] excluded patients who received NAC before RNU.

Several publications provided additional analyses of specific subset of patients extracted from the main cohorts. Lee J.K. et al. stratified patients by primary tumor location (separate analyses for renal pelvic tumors (RPT) and ureteral tumors (UT)) [10]. Zhao et al. conducted additional analyses for patients stratified by the presence of hydronephrosis at the time of diagnosis [15]. Sundi et al. provided separate outcome analysis for patients who did not receive NAC [12]. Waldert et al. and Xia et al. separately analyzed a muscle-invasive cohort (patients with ≥pT2 tumors) or a “higher-risk” cohort (patients with ≥pT2 and/or ≥G3 tumors), respectively [13,14].

Only two studies reported the reasons for RNU delay. In Sundi et al.’s study, 50% of delayed RNU were caused by an administration of NAC, while an additional 17% were delayed because of the initial endoscopic management [12]. Performance of URS before RNU was the main cause of delay in the study by Nison et al. [11].

### 3.3. Risk of Bias (RoB) and Quality Assessment of Included Studies

The evaluation of RoB and confounding assessment for included studies is shown in Figure 2. Due to the retrospective design, all selected articles carried a high RoB. The issue of confounding was addressed by most studies, as statistical adjustment was performed in five out of seven articles through multivariate analyses [9,10,12,14,15]. Of them, all were adjusted for pathological tumor stage and grade. However, other confounders were not uniformly taken into account.

### 3.4. Definition of Deferred Radical Nephroureterectomy

Surgical wait time was predominantly defined as the interval between initial imaging diagnosis and radical surgery of UTUC. Significant heterogeneity in the definition of “deferred RNU” was found across the included studies. Three reports (Lee H.Y. et al., Sundi et al., Waldert et al.) used a cut-off of 90 days (3 months) [9,12,13], while a single study (Lee J.N. et al.) used a cut-off of 30.5 days (1 month) [10]. Nison et al. used the following delay intervals: ≤30 days (<1 month), 31–60 days (1–2 months), 61–90 days (2–3 months), >90 days (>3 months) [11]. Zhao et al. presented groups categorized by the following time intervals: ≤30 days (<1 month), 31–90 days (1–3 months), >90 days (>3 months) [15]. Xia et al. divided patients into those who underwent RNU: 1–7 days, 8–30 days, 31–60 days (1–2 months), 61–90 days (2–3 month), 91–120 days (3–4 months), and 121–180 days (4–6 months) after UTUC diagnosis [14].

### 3.5. Results of Systematic Review (Qualitative Synthesis)

#### 3.5.1. Cancer-Specific Survival (CSS)

Data regarding CSS were reported in five out of seven studies (Table 3) [9,10,12,13,15]. Of them, three found a significant impact of the delay in RNU on CSS in the overall cohort or a subset of patients [10,13,15].

Lee J.N. et al. observed no significant difference in CSS between patients who underwent RNU ≤ 30 days or >30 days after UTUC diagnosis [10]. However, subgroup analysis of patients with UT revealed that CSS was significantly improved in patients who had RNU within 30 days (5-year CSS: 87.9% vs. 54.5%, *p <* 0.001). Multivariable Cox regression analysis confirmed that a surgical wait time of more than 1 month was one of the independent prognostic factors of worse CSS in a subset of patients with UT (HR = 6.26, 95% CI: 1.90–20.62, *p =* 0.003). However, no association was found in a subset of patients with RPT [10]. Using univariable Cox regression analysis, Nison et al. found no significant differences in CSS for any reported time interval, even in a subset of patients with confirmed muscle-invasive UTUC [11]. Sundi et al. demonstrated no significant differences between the “early” (≤90 days) and “delayed” (>90 days) RNU groups with respect to 5-year CSS rates in the overall cohort (71.6% vs. 70.6%, *p >* 0.05), as well as in a subset of patients who did not receive NAC (71.6% vs. 81.5%, *p >* 0.05) [12]. Waldert et al. showed no significant difference in CSS between patients who had RNU at ≤ 90 days or > 90 days after UTUC diagnosis (5-year CSS: 72% vs. 63%, *p =* 0.153) [13]. In univariate Cox regression analysis, the time from diagnosis to RNU (as a continuous variable) was associated with worse CSS in a subset of patients with muscle-invasive disease (HR = 1.005, 95% CI: 1.001–1.010, *p =* 0.003); this was not true for the whole cohort (*p =* 0.658) [13]. Zhao et al. noted no significant difference in CSS for patients undergoing RNU at 31–90 days compared with ≤30 days; however, those with a delay > 90 days had a significantly worse CSS (65.8% vs. 70.9% vs. 39.6%, *p =* 0.032) [15]. In multivariate Cox regression analysis, performed for a subset of patients with hydronephrosis at the time of UTUC diagnosis, surgical wait time > 60 days was one of the independent risk factors for worse CSS (HR = 1.74, 95% CI: 1.07–2.82, *p =* 0.026) [15].

The initially planned meta-analysis for CSS was not possible because of the paucity and heterogeneity of available data.

#### 3.5.2. Overall Survival (OS)

Data regarding OS were reported in four out of seven studies (Table 3) [9,12,14,15]. Three of them found a significant impact of delay in RNU on OS in the overall cohort or a subset of patients [9,14,15].

Lee H.Y. et al. showed that an “early” (≤90 days) RNU group had a better 5-year OS rate, compared to a “delayed” (>90 days) RNU group (72.9% vs. 63.5%, *p =* 0.015) [9]. In addition, on multivariate Cox regression analysis, RNU after 90 days was associated with a significantly worse OS (HR = 1.55, 95% CI: 1.03–2.33, *p =* 0.035) [9]. Conversely, Sundi et al. found no significant difference in OS between patients undergoing RNU at ≤90 days and >90 days from UTUC diagnosis [12]. Xia et al. demonstrated that patients with RNU delay time of 31–60 days, 61–90 days, and 91–120 days had similar OS compared with patients who had a delay in RNU of 8—30 days in both the overall cohort and “higher-risk” cohort (≥pT2 and/or ≥G3 tumors) [14]. However, patients with RNU deferred for 121–180 days had worse OS in both overall (HR = 1.61, 95% CI: 1.19–2.19, *p =* 0.002) and “higher-risk” (≥pT2 and/or ≥G3 tumors; HR = 1.56, 95% CI: 1.11–2.20, *p =* 0.01) cohorts, respectively [14]. Zhao et al. showed no significant difference in OS for patients undergoing RNU at 31–90 days, compared with ≤30 days. However, those with a delay >90 days had worse OS (56.4% vs. 59.3% vs. 35.1%, *p =* 0.045) [15]. On multivariate Cox regression analysis of patients with hydronephrosis at the time of diagnosis, surgical wait time > 60 days was one of the independent risk factors for worse OS (HR = 2.05, 95% CI: 1.20–3.50, *p =* 0.009) [15].

A forest plot comparing OS between “long” and “short” surgical waiting time groups is provided in Appendix A.

#### 3.5.3. Recurrence-Free Survival (RFS)

Data regarding RFS were reported in three out of seven studies [10,11,13]. Of them, one found a significant impact of the delay in RNU on RFS in the overall cohort or a subset of patients [10].

Five-year RFS rates reported by Lee J.N. et al. were comparable between patients who underwent RNU ≤ 30 days or >30 days after UTUC diagnosis (77.6% vs. 73.9%, *p =* 0.534) [10]. In a subset of patients with RPT, delay in RNU > 30 days was associated with improved 5-year RFS rate (66.3% vs. 91.6%, *p =* 0.028). However, it was not confirmed in the univariable Cox regression analysis (*p =* 0.537). In a subgroup analysis including patients with UT, the delay in RNU > 30 days was associated with significantly worse 5-year RFS (85.6% vs. 60.7%, *p =* 0.007) and was one of the independent prognostic factors for worse RFS in multivariable Cox regression analysis (HR = 4.120, 95% CI: 1.38–12.30, *p =* 0.011) [10]. Nison et al. found no significant difference in RFS for any reported time interval (delay < 1 month as a reference, delay > 3 months: HR = 0.96, 95% CI: 0.70–1.29, *p =* 0.78) [11]. Furthermore, in another study from Waldert et al., patients who underwent RNU > 90 days after UTUC diagnosis had similar 5-year RFS compared to those who underwent RNU ≤ 90 days (68% vs. 51%, *p =* 0.066) [13].

The initially planned meta-analysis for RFS was not possible because of the paucity and heterogeneity of available data.

## 4. Discussion

In the present systematic review, we conducted a qualitative synthesis of current data regarding the impact of delaying RNU on long-term oncological outcomes in patients with UTUC. According to the current evidence, long surgical waiting time for RNU (especially beyond 3 months after UTUC diagnosis) could be considered as an important risk factor having a negative impact on survival parameters. Notably, the “safe window” for RNU seems to be shorter specifically for high-risk patients such as those diagnosed with UT or hydronephrosis.

Diagnosis of UTUC and proper preoperative determination of the disease stage and grade can often be challenging. As it is a crucial step in terms of planning the treatment (conservative management vs. RNU), diagnostic URS with biopsy is a valuable tool in case of inconclusive computed tomography urography (CTU) findings. Even though URS may clearly increase the time between diagnosis and treatment, it is rarely associated with a long delay (e.g., more than 3 months). In a single study included in the present systematic review, patients who underwent URS and delayed RNU had similar CSS and RFS compared to the patients who underwent RNU without previous URS [11]. Nonetheless, the results of a recent meta-analysis including 16 retrospective series confirmed that URS with biopsy followed by RNU could be associated with significantly worse intravesical RFS (but not with CSS, OS, and metastasis-free-survival), compared to RNU alone [16]. These findings could be explained by increased risk of tumor seeding during endoscopic biopsy or the manipulation of the ureteroscope. Therefore, URS (particularly with biopsy) seems reasonable only in uncertain diagnostic cases, when no NAC is planned and the disease cannot be classified as high-risk based on other clinical factors, such as tumor size, multifocality (based on CTU results) or high-grade cytology results.

Currently, KSS (e.g., endoscopic ablation, segmental ureteral resection) is the preferred approach in low-risk UTUC. Gadzinski et al. showed that the delay in RNU related to previous KSS did not affect survival outcomes in patients with UTUC [17]. No specific cut-off for delay interval was reported and the study included a relatively small sample size (n = 73). Authors reported comparable 5-year OS (64% vs. 59%) and 5-year CSS (91% vs. 80%) between patients in the delayed RNU group (with previous conservative treatment) and immediate RNU group (without previous conservative treatment). However, a significant pathologic progression was observed in 43% of the cases in the delayed surgical group, when compared to the initial endoscopic pathology. In another multi-institutional retrospective study, Gurbuz et al. confirmed that endoscopic ablation prior to RNU was not associated with decreased CSS and disease-free survival (DFS) [18]. This evidence suggests that delayed RNU preceded by KSS could be a feasible option after endoscopic management failure; however, proper patient selection for initial KSS seems to be the key to guaranteeing satisfactory oncological outcomes [17,18].

Ureteral location is considered as an important negative prognostic factor in patients with UTUC. Recent meta-analysis including 10,537 patients with RPT and 6299 patients with UT demonstrated that ureteral location of UTUC is associated with decreased CSS, OS, and DFS [19]. More aggressive behavior of UT, potentially related to tumor’s surrounding environment (e.g., thin periureteral layer of muscular and fatty tissue, compared to renal parenchyma), raises the question about the safe delay interval in radical treatment. Based on the results of Lee et al.’s study, a surgical wait time of more than 1 month after UTUC diagnosis might be associated with significantly worse prognosis in patients with UT [10]. In addition, a shorter “safe window” for radical treatment was noted by Zhao et al. for patients with UTUC presenting hydronephrosis at the time of diagnosis [15]. In this cohort, the CSS and OS of the patients with surgical wait time of more than 60 days were significantly lower than those of patients with surgical wait time of less than 60 days. To support their results, the authors hypothesized that increased pressure of the renal pelvis and ureter due to hydronephrosis may lead to easier peripheral invasion or ischemic changes in surrounding tissues, inducing the expression of hypoxia-inducible factors involved in tumor growth [15]. Moreover, a variety of independent factors can influence oncological outcomes, regardless of surgical waiting time. Pathological stage, grade, LNI, LVI, positive surgical margin, presence of tumor necrosis, hydronephrosis, and tumor size were also associated with worse CSS and OS in several selected studies [10,11,12,15]. OS was negatively influenced by pathological tumor grade, stage, size, multifocality, LNI and hydronephrosis [9,15]. Therefore, surgical waiting time should not be considered as the sole independent risk factor of worse oncological outcomes in patients with UTUC. In view of the abovementioned evidence, determining the safe delay of the RNU should be conducted according to the individual case risk profile, based on all available clinical factors. Nevertheless, further studies are required to make strict recommendations.

It needs to be emphasized that the delay in RNU does not always delay the treatment. There is growing evidence that cisplatin-based NAC can lead to a significant downstaging or a complete response on final pathologic examination of the RNU specimen (resulting in CSS and OS improvement), which is why NAC is increasingly utilized in the management of UTUC [20]. On the other hand, NAC might delay surgical treatment of UTUC, potentially leading to disease progression in chemo-resistant patients [21]. In addition, patients undergoing NAC may suffer from toxicities related to chemotherapy, which may delay surgery even further [21]. Thus, development and validation of preoperative models are extremely important as the scope of future research, in order to guide selection of the most suitable patients with UTUC who will benefit from NAC. Unfortunately, selected papers did not include subgroup analyses of patients receiving NAC before RNU. [11]. Only one study by Sundi et al. addressed this issue and demonstrated no significant differences in CSS between the “early” (≤90 days) and “delayed” (>90 days) RNU groups in a total cohort (50% of patients receiving NAC) and subgroup of patients not receiving NAC. Therefore, due to paucity of data, the safe delay in RNU in patients receiving NAC could not be reliably established.

The delay in RNU can be caused by a number of reasons, both disease and patient-related. In some analyzed studies longer waiting time was mainly caused by NAC and URS prior to the surgery [11,12]; however, several studies did not provide specific causes of RNU delay. Potential reasons for the delay, such as limited surgical schedules, delayed referral to urologist due to high burden on the health systems, contraindications to surgery, patients’ attitude should be considered as important factors that occur in clinical practice. Delayed surgical wait times have an unfavorable impact on the overall quality of life and psychological comfort of the patients. Various studies confirmed that long waiting for surgery aggravates anxiety and psychological distress in patients with various urologic neoplasms [22,23]. The delay can also influence the patients’ close relatives, increasing stress and creating frustration. What is more, the psychological well-being of patients is crucial in postoperative compliance and maintaining a positive relationship with the physicians. That is why mental health can influence the oncological outcomes in patients with UTUC and should not be underestimated.

Although the delay of a radical treatment in patients with UTUC seems to be safe and acceptable up to 3 months, it needs to be emphasized that the current data are not sufficient to reliably consider this as strong evidence. There are many potential causes of delay in RNU that could occur in clinical practice. Thus, the delay of a definitive treatment in patients with UTUC should be done with caution and rational basis in each individual case. On the basis of our synthesized data, we recommend further studies to prospectively assess the association between RNU delay and oncological outcomes in UTUC patients. Future studies should include homogenous populations in terms of causes of RNU delay (e.g., NAC administration or URS procedure before RNU) or provide detailed subgroup analyses. This could help elucidate the oncologic impact of particular delays and prepare for future unexpected events that could result in prolonged delays in definitive care in patients with UTUC.

Several limitations of the present work should be mentioned. The first and most important limitation is the retrospective and heterogeneous nature of included studies. Second, most articles did not report the reason for RNU delay. Due to possible selection bias, elderly patients with more comorbidities could be more likely to be selected for the delayed RNU than younger patients without comorbidities (possible attrition bias). Third, the studies included in our paper were conducted in different geographical regions and observed differences in the results might reflect regional ethnic differences. Fourth, as highlighted by their large CIs and small sample size, some studies might be underpowered to detect a difference in oncological outcomes between analyzed delay intervals. Fifth, the reasons of longer waiting time were not reported in some of the selected studies, thus, results of this study may not be applicable for specific subsets of patients (e.g., receiving NAC, patients with <pT2 tumors). Finally, the planned meta-analysis was not possible because of the heterogeneity of available data.

## 5. Conclusions

According to the current evidence, long surgical waiting time for RNU (especially more than 3 months after UTUC diagnosis) could be considered as an important risk factor having a negative impact on oncological outcomes in patients with UTUC; however, the results of the particular studies are still inconsistent. The safe delay for RNU might be shorter in specific subsets of high-risk patients, such as those with UT and/or hydronephrosis at the time of diagnosis. Nonetheless, high-quality additional studies are required to establish evidence for valid recommendations.

## Figures and Tables

**Figure 1 jcm-11-04007-f001:**
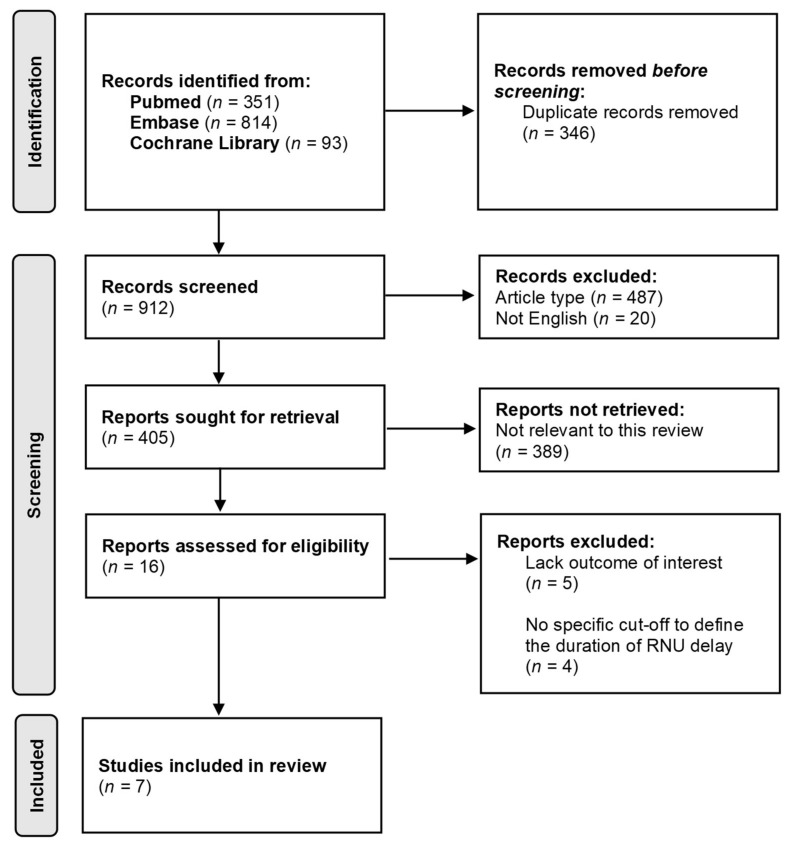
A PRISMA flowchart describing the study selection process. RNU = radical nephroureterectomy.

**Figure 2 jcm-11-04007-f002:**
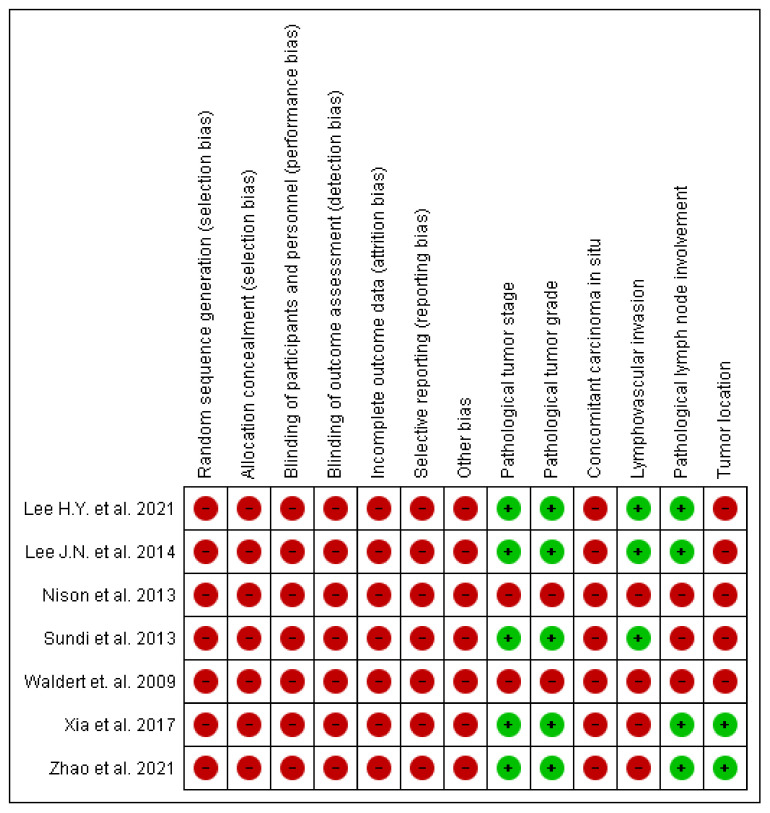
The risk of bias and confounding assessment for included studies [9,10,11,12,13,14,15].

**Table 1 jcm-11-04007-t001:** Baseline characteristics of included studies.

First Author, Year [Reference]	Journal	Geographical Region	Study Type	Study Duration	Number of Patients	Definition of RNU Delay	Median Time to RNU, Days	Follow up, Months	Reported Outcomes
Lee H.Y. et al., 2021 [9]	*Urologic Oncology: Seminars and Original Investigations*	Asia	Retrospective Multi-center	2000–2019	665	Group 1: ≤90 days Group 2: >90 days	NR	Group 1: mean 52.3 Group 2: mean 34.2	OS
Lee J.N. et al., 2014 [10]	*Journal of Surgical Oncology*	Asia	Retrospective Single-center	2001–2010	138	Group 1: ≤30 days Group 2: >30 days	Mean: 16.6 Mean: 70.1	All patients: median 40	CSS, RFS
Nison et al., 2013 [11]	*World Journal of Urology*	Europe	Retrospective Multi-center	1995–2011	512	Group 1: ≤30 days Group 2: 31–60 days Group 3: 61–90 days Group 4: >90 days	NR	All patients: median 23.6	CSS, RFS
Sundi et al., 2013 [12]	*Urological Oncology*	North America	Retrospective Single-center	1990–2007	240	Group 1: ≤90 days Group 2: >90 days	Mean: 24 Mean: 432	All patients: median 29	CSS, OS
Waldert et al., 2009 [13]	*BJU International*	Multinational	Retrospective Multi-center	2000–2007	187	Group 1: ≤90 days Group 2: >90 days	Median: 33 Median: 110	All patients: median 47.5	CSS, RFS
Xia et al., 2017 [14]	*Urologic Oncology: Seminars and Original Investigations*	North America	Retrospective Population-based registry	2004–2013	3581	Group 1: 8–30 days Group 2: 1–7 days Group 3: 31–60 days Group 4: 61–90 days Group 5: 91–120 days Group 6: 121–180 days	NR	All patients: median 40.4	OS
Zhao et al., 2021 [15]	*Frontiers in Oncology*	Asia	Retrospective Single-center	2008–2019	316	Group 1: ≤30 days Group 2: 31–90 days Group 3: >90 days	Median: 12 Median: 42 Median 191	All patients: median 43	OS, CSS

Abbreviations: CSS = cancer-specific survival; NR = not reported; OS = overall survival; RFS = recurrence-free survival; RNU = radical nephroureterectomy.

**Table 2 jcm-11-04007-t002:** Clinical and pathological characteristics of main cohorts in included studies.

First Author, Year [Reference]	Gender, n (%)	Preoperative Hydronephrosis, n (%)	URS, n (%)	Tumor Location, n (%)	RNU Approach, n (%)	Pathological Tumor stage, n (%)	Pathological Tumor grade, n (%)	LNI, n (%)	Concomitant CIS, n (%)	LVI, n (%)	AC, n (%)
Lee H.Y. et al., 2021 [9]	Male: 297 (49.5) Female: 303 (50.5)	NR	Yes: 491 (74.0) No: 174 (26.0)	NR	NR	<T2: 198 (33.0) ≥T2: 361 (67.0)	G1: 77 (14.8) G2: 62 (11.9) G3: 381 (73.3)	Yes: 44 (7.3) No ^a^: 556 (92.7)	Yes: 20 (3.3) No: 580 (96.7)	Yes: 133 (22.2) No: 467 (77.8)	Yes: 89 (14.8) No: 511 (85.2)
Lee J.N. et al., 2014 [10]	Male: 96 (69.6) Female: 42 (30.4)	Yes: 100 (72.5) No: 38 (27.5)	NR	RPT: 58 (42.0) UT: 80 (58.0)	Open: 36 (26.1) Laparoscopic: 102 (73.9)	<T2: 50 (36.2) ≥T2: 88 (63.8)	LG: 46 (33.3) HG: 92 (66.7)	Yes: 10 (7.2) No ^a^: 128 (92.8)	Yes: 7 (5.1) No: 131 (94.9)	Yes: 27 (19.6) No: 111 (80.4)	Yes: 43 (31.2) No: 95 (68.8)
Nison et al., 2013 [11]	Male: 348 (68.0) Female: 164 (32.0)	NR	Yes: 170 (33.2) No: 342 (66.8)	RPT: 277 (54.1) UT: 172 (33.6) Multifocal: 63 (12.3)	NR	<T2: 252 (49.2) ≥T2: 260 (50.8)	G1: 62 (12.1) G2: 154 (30.1) G3: 296 (57.8)	Yes: 39 (7.6) No ^a^: 473 (92.4)	NR	Yes: 126 (24.6) No: 368 (75.4)	NR
Sundi et al., 2013 [12]	Male: 157 (65.4) Female: 83 (34.6)	NR	NR	RPT: 140 (58.3) UT: 100 (41.7)	NR	<T2: 120 (50.0) ≥T2: 120 (50.0)	LG: 51 (21.2) HG: 189 (78.8)	Yes: 30 (12.5) No ^a^: 210 (87.5)	Yes: 101 (42.1) No: 139 (57.9)	Yes: 74 (30.8) No: 166 (69.2)	Yes: 38 (15.8) No: 202 (84.2)
Waldert et al., 2009 [13]	Male: 150 (80.2) Female: 37 (19.8)	NR	Yes: 49 (26.2) No: 138 (73.8)	RPT: 88 (47.1) UT: 99 (52.9)	Open: 151 (80.7) Laparoscopic: 36 (19.3)	<T2: 97 (51.9) ≥T2: 90 (48.1)	LG: 62 (33.2) HG: 125 (66.8)	Yes: 17 (9.1) No ^a^: 170 (90.9)	Yes: 78 (41.7) No: 109 (58.3)	Yes: 54 (28.9) No: 133 (71.1)	Yes: 30 (16.0) No: 157 (84.0)
Xia et al., 2017 [14]	Male: 2038 (56.9) Female: 1543 (43.1)	NR	NR	RPT: 2428 (67.8) UT: 1153 (32.2)	NR	<T2: 1865 (52.1) ≥T2: 1429 (41.7)	G1-2: 1273 (35.6) G3-4: 2308 (64.4)	Yes: 147 (4.1) No ^a^: 3434 (95.9)	NR	NR	Yes: 357 (10.0) No: 3224 (90.0)
Zhao et al., 2021 [15]	Male: 205 (64.9) Female: 111 (35.1)	Yes: 158 (50.0) No: 158 (50.0)	NR	RPT: 173 (54.7) UT: 143 (45.3)	Open: 67 (21.2) Laparoscopic: 249 (78.8)	<T2: 87 (27.5) ≥T2: 229 (72.5)	LG: 81 (25.6) HG: 234 (74.4)	Yes: 34 (10.8) No ^a^: 282 (89.2)	NR	Yes: 38 (12.0) No: 278 (88.0)	Yes: 32 (10.1) No: 284 (89.9)

^a^ pathological N0 and/or Nx. Abbreviations: AC = adjuvant chemotherapy; CIS = carcinoma in situ; HG = high grade; LG = low grade; LNI = lymph node invasion; LVI = lymphovascular invasion; NR = not reported; RNU = radical nephroureterectomy; RPT = renal pelvic tumor; URS = ureteroscopy; UT = ureteral tumor.

**Table 3 jcm-11-04007-t003:** Primary oncological outcomes of interest reported in included studies.

First Author, Year [Reference]	Subgroup	5-Year CSS	*p*-Value	5-Year OS	*p*-Value	Multivariable Cox Regression Analysis—CSS HR [95% CI]	*p*-Value	Multivariable Cox Regression Analysis—OS HR [95% CI]	*p*-Value
Lee H.Y. et al., 2021 [9]	NA	NR	NR	Delay ≤ 90 days: 72.9% Delay > 90 days: 63.5%	0.015 *	NR	NR	Delay ≤ 90 days: Ref Delay > 90 days: 1.55 [1.03–2.33]	0.035 *
Lee J.N. et al., 2014 [10]	All patients	Delay ≤ 30 days: 77.3% Delay > 30 days: 69.1%	0.087	NR	NR	NR	*Univariable* 0.089	NR	NR
	Renal pelvis tumors	Delay ≤ 30 days: 63.9% Delay > 30 days: 90.1%	0.084	NR	NR	NR	*Univariable* 0.085	NR	NR
	Ureteral tumors	Delay ≤ 30 days: 87.9% Delay > 30 days: 54.5%	<0.001 *	NR	NR	Delay ≤ 30 days: ref Delay > 30 days: 6.26 [1.90–20.62]	0.003 *	NR	NR
Nison et al., 2013 [11]	NA	NR	NR	NR	NR	*Univariable* Delay ≤ 30 days: ref Delay 31–60 days: 1.00 [0.50–1.98] Delay 61–90 days: 0.84 [0.37–1.91] Delay > 90 days: 0.92 [0.45–1.89]	- 0.99 0.68 0.68	NR	NR
Sundi et al., 2013 [12]	All patients	Delay ≤ 90 days: 71.6% Delay > 90 days: 70.6%	NS	Delay ≤ 90 days: 61.3% Delay > 90 days: 77.0%	NS	NR	NR	Delay ≤ 90 days: ref Delay > 90 days: 1.54 [0.73–3.25]	0.25
	Patients not receiving NAC	Delay ≤ 90 days: 71.6% Delay > 90 days: 81.5%	NS	Delay ≤ 90 days: 61.3% Delay > 90 days: 77%	NS	NR	NR	Delay ≤ 90 days: ref Delay > 90 days: 0.94 [0.28–3.08]	0.92
Waldert et al., 2009 [13]	All patients	Delay ≤ 90 days: 72% Delay > 90 days: 63%	NS	NR	NR	*Univariable* Time as continuous variable: 1.00 [0.99–1.01]	0.658	NR	NR
	≥pT2 on RNU	Delay ≤ 90 days: 49% Delay > 90 days: 45%	NS	NR	NR	Time as continuous variable: 1.005 [1.001–1.010]	0.03	NR	NR
Xia et al., 2017 [14]	All patients	NR	NR	Delay 8–30 days: 64.2% Delay 1–7 days: 58.5% Delay 31–60 days: 61.8% Delay 61–90 days: 60.6% Delay 91–120 days: 61.5% Delay 121–180 days: 36.6%	*	NR	NR	Delay 8–30 days: ref Delay 1–7 days: 1.32 [1.06–1.67] Delay 31–60 days: 1.11 [0.97–1.27] Delay 61–90 days: 1.09 [0.91–1.30] Delay 91–120 days: 1.00 [0.74–1.35] Delay 121–180 days: 1.61 [1.19–2.19]	- 0.016 * 0.126 0.360 0.976 0.002 *
	≥pT2 and/or ≥G3 on RNU	NR	NR	Delay 8–30 days: 57.2% Delay 1–7 days: 55.8% Delay 31–60 days: 53.5% Delay 61–90 days: 51.6% Delay 91–120 days: 51.6% Delay 121–180 days: 26.5%	*	NR	NR	Delay 8–30 days: ref Delay 1–7 days: 1.24 [0.95–1.61] Delay 31–60 days: 1.10 [0.94–1.27] Delay 61–90 days: 1.07 [0.88–1.31] Delay 91–120 days: 0.94 [0.66–1.34] Delay 121–180 days: 1.56 [1.11–2.20]	- 0.114 0.231 0.510 0.744 0.010 *
Zhao et al., 2021 [15]	All patients	Delay ≤ 30 days: 65.8% Delay 31–90 days: 70.9% Delay > 90 days: 39.6%	0.032 *	Delay ≤ 30 days: 56.4% Delay 31–90 days: 59.3% Delay > 90 days: 35.1%	0.045 *	NR	NR	NR	NR
	Patients with hydronephrosis	Delay ≤ 60 days: 61.7% Delay > 60 days: 49.1%	0.041 *	Delay ≤ 60 days: 55.1% Delay > 60 days: 44.2%	0.023 *	Delay ≤ 60 days: ref Delay > 60 days: 1.74 [1.07–2.82]	0.026 *	Delay ≤ 60 days: ref Delay > 60 days: 2.05 [1.20–3.50]	0.009 *

* Statistically significant *p*-value. Abbreviations: CI = confidence interval; CSS = cancer-specific survival; HR = hazard ratio; OS = overall survival; NA = not applicable; NR = not reported; NS = not statistically significant; RNU = radical nephroureterectomy.

## Data Availability

Not applicable.

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
