# Peer review of "The Impact of Surgical Waiting Time on Oncological Outcomes in Patients with Upper Tract Urothelial Carcinoma Undergoing Radical Nephroureterectomy: A Systematic Review"

_jcm, 2022, doi:10.3390/jcm11144007_

Round 1
Reviewer 1 Report
The authors conducted a systematic review on timing of radical nephroureterectomy (RNU) for upper tract urothelial carcinoma (UTUC). They reported a metanalysis data on effect of delayed RNU on cancer-specific survival (CSS), overall survival (OS), and recurrence-free survival (RFS). They found performance of RNU more than 3 months after UTUC diagnosis may be associated with significantly worse oncological outcomes, however, results of particular studies are still inconsistent. The safe delay for RNU might be shorter in specific subsets of patients, such as cases with ureteral tumor or hydronephrosis at the time of diagnosis.
Major comments:
1- How did the authors define time to RNU (from initial diagnostic imaging or biopsy)?
2- The authors need to stratify the results based on whether patients receive NAC, as delaying due to NAC could impose selection bias.
3- Does delaying RNU have similar effect in patients with
4- The lack of data regarding adjuvant treatments makes the result of this study not generalizable.
5- The authors should consider this paper as a hypothesis producing study and discuss design of future studies to investigate the hypothesis.
Author Response
We highly appreciate Reviewer’s feedback and valuable comments. A detailed report on the amendments is presented below.
- How did the authors define time to RNU (from initial diagnostic imaging or biopsy)?
Our response: Surgical waiting time was predominantly defined as the interval between initial imaging diagnosis and radical surgery of UTUC. We added this information (Line 188 – 189).
- The authors need to stratify the results based on whether patients receive NAC, as delaying due to NAC could impose selection bias.
Our response: We totally agree with Reviewer that stratification of patients based on NAC administration would be valuable. However, the included studies did not provide data for subgroup of patients receiving NAC. Only Sundi et al. addressed the issue of NAC administration in the context of RNU delay (Line 213 – 216). Therefore, we could not reliably synthetize data based on NAC administration. We additionally stated this in the last paragraph of discussion (limitations) (Line 396 – 398).
- Does delaying RNU have similar effect in patients with
Our response: None of the selected articles provided separated analyses for patients with
- The lack of data regarding adjuvant treatments makes the result of this study not generalizable.
Our response: As above, the inability to perform data synthesis for specific subsets of patients was additionally stated as limitation of this study (Line 396 – 398).
- The authors should consider this paper as a hypothesis producing study and discuss design of future studies to investigate the hypothesis.
Our response: We additionally discussed the need for future studies analyzing the impact of RNU delay on oncological outcomes (Line 380 – 385).
Reviewer 2 Report
This study investigated the impact of surgical waiting time on oncological outcomes in patients with upper tract urothelial carcinoma (UTUC) undergoing radical nephroureterectomy (RNU) by the systematic review. You concluded that the performance of RNU more than 3 months after UTUC diagnosis may be associated with significantly worse oncological outcomes, although further additional studies are required to establish clear and reliable recommendations.
This study is straightforward, but does not change clinical practice. Additionally, we need to be careful in interpreting your conclusions.
There are some major points to propound:
1) You reviewed the 7 eligible studies enrolling 5639 patients. Some studies which were reviewed have concluded that the surgical waiting time has no effect cancer-specific survival (CSS) and overall survival (OS) in patients with UTUC undergoing radical RNU. You should create forest plots comparing CSS and OS between long surgical waiting time group and short surgical waiting time group, in order to improve the quality of your conclusions.
2) It is unlikely that delayed surgical wait period of only a few months would affect the oncological outcome, and your results could be misleading to readers. It would be advisable to review the impact of surgical wait time on oncologic outcomes, including the impact of other factors such as time to diagnosis, tumor stage, and tumor location.
Author Response
We highly appreciate Reviewer’s feedback and valuable comments. A detailed report on the amendments is presented below.
- You reviewed the 7 eligible studies enrolling 5639 patients. Some studies which were reviewed have concluded that the surgical waiting time has no effect cancer-specific survival (CSS) and overall survival (OS) in patients with UTUC undergoing radical RNU. You should create forest plots comparing CSS and OS between long surgical waiting time group and short surgical waiting time group, in order to improve the quality of your conclusions.
Our response: As it was stated in the last paragraph of discussion section, the initially planned meta-analysis was not possible because of the heterogeneity of available data (mainly due to various reference time intervals in multivariable or univariable analyses across selected studies). It mainly applies to cancer-specific survival (CSS) and recurrence-free survival (RFS) analyses. However, a supplementary forest plot could be created for overall-survival (OS) data – we provided it in supplementary file and added an appropriate statement in 3.5.2 section (Line 250 - 251). However, it is rather for overview purposes – studies by Zhao and Xia reports different reference time intervals compared to Lee HY and Sundi. Also, we added additional statements that creating a reliable forest plots of pooled hazard ratios was not possible for CSS and RFS (Line 228 – 229, 269 – 270). Thus, our study was primarily designed as a systematic review.
- It is unlikely that delayed surgical wait period of only a few months would affect the oncological outcome, and your results could be misleading to readers. It would be advisable to review the impact of surgical wait time on oncologic outcomes, including the impact of other factors such as time to diagnosis, tumor stage, and tumor location.
Our response: According to EAU guidelines, factors such pT2 and higher tumor stage, positive LNI or LVI are are well-established predictors of survival in patients with UTUC. However, our aim was to synthetize the data regarding RNU delay. We added an additional discussion section regarding prognostic factors in patients with UTUC (Line 334 – 344) and changed the conclusions (indicating thar RNU delay should be considered as one of the significant risk factor of worse survival) (Line 401 – 404).
Reviewer 3 Report
The authors described the impacts of surgical waiting time on oncological outcomes in UTUC patients who received RNU. The concept of the current study is interesting and has clinical value. However, there are some points to be improved before publication. Some comments are described below:
1. Does the Pathological tumorical stage, LNI or LVI associate with OS, CSS or RFS? If so, these data would be “risk factors” and impact outcomes regardless of waiting time.
2. As author mentioned “All studies except one excluded patients who received NAC before RNU”. In my opinion, this could be good data to analyze. Waiting time with NAC might show significance on the outcome compared to without NAC. More details and discussions about “delay reason” could be shown.
Author Response
We highly appreciate Reviewer’s feedback and valuable comments. A detailed report on the amendments is presented below.
- Does the Pathological tumorical stage, LNI or LVI associate with OS, CSS or RFS? If so, these data would be “risk factors” and impact outcomes regardless of waiting time.
Our response: According to Figure 2, tumor stage, LNI or LVI (and other factors such as tumor grade) were confounders in multivariable analyses reported in majority of selected articles. According to EAU guidelines, these are well-established predictors of survival in patients with UTUC. Obviously, high tumor stage, positive LNI or LVI were associated with worse oncological outcomes in almost all multivariable analyses, however, our aim was to synthetize the data regarding RNU delay. We totally agree that additional risk factors (like above mentioned tumor-related factors) have an impact the oncological outcomes in patients with UTUC, and RNU delay should not be considered as sole and only risk factor. We added an additional discussion section and changed the conclusions (indicating thar RNU delay should be considered as one of the significant risk factor of worse survival) (Line 334 – 344, Line 401 – 404).
- As author mentioned “All studies except one excluded patients who received NAC before RNU”. In my opinion, this could be good data to analyze. Waiting time with NAC might show significance on the outcome compared to without NAC. More details and discussions about “delay reason” could be shown.
Our response: Unfortunately, the included studies did not provide data for subgroup of patients receiving NAC. Only Sundi et al. addressed the issue of NAC in the context of RNU delay (Line 213 – 216). Therefore, we could not reliably synthetize data based on NAC administration. We additionally stated this in the last paragraph of discussion (Line 396 – 398). Delay reasons were additionally discussed (Line 361 – 366).
Round 2
Reviewer 2 Report
The author appropriately responded to the reviewers' comments. The quality of this paper has improved.